# The Application of Cellulose Acetate Membranes for Separation of Fermentation Broths by the Reverse Osmosis: A Feasibility Study

**DOI:** 10.3390/ijms231911738

**Published:** 2022-10-03

**Authors:** Wirginia Tomczak, Marek Gryta

**Affiliations:** 1Faculty of Chemical Technology and Engineering, Bydgoszcz University of Science and Technology, 3 Seminaryjna Street, 85-326 Bydgoszcz, Poland; 2Faculty of Chemical Technology and Engineering, West Pomeranian University of Technology in Szczecin, Piastów Ave. 42, 71-065 Szczecin, Poland

**Keywords:** 1,3-propanediol, cellulose acetate, fermentation broth, fouling, glycerol, osmotic pressure, rejection, reverse osmosis, separation, size effect

## Abstract

Recently, there has been a special research focus on the bioconversion of glycerol to 1,3-propanediol (1,3-PD) due to its significance in the chemical industry. However, the treatment and separation of fermentation broths is a great challenge. Currently, the reverse osmosis (RO) process is a reliable state-of-the-art technique for separation of biological solutions. This study (as the first to do so) investigated the feasibility of separation of 1,3-PD broths with the use of cellulose acetate (CA) membrane by the RO process. The experiments were carried out using the installation equipped with the plate module, under the transmembrane pressure (TMP) and temperature of 1 MPa and 298 K, respectively. It was found that the used membrane was suitable for broth separation. Indeed, it was noted that 1,3-PD, as a target product, migrated through the membrane; meanwhile, other broth components were rejected in various degrees. Moreover, it was proven that retention of carboxylic acids tended to increase with increasing molecular weight, according to the following order: succinic acid > lactic acid > acetic acid > formic acid. With regards to ions, retention degree increased with the increase of ionic radius and decrease of diffusion coefficient. Finally, it was demonstrated that the CA membrane is resistant to irreversible fouling, which has a positive effect on the economic viability of the process.

## 1. Introduction

Currently, there is a continuously increasing worldwide concern about the production of high added-value bioproducts from renewable feedstocks by green processes. Consequently, many researchers have made remarkable achievements in obtaining 1,3-propanediol (1,3-PD) via the glycerol fermentation process with the use of various bacterial strains [1,2,3,4,5]. However, effective separation of target products from post-fermentation solutions remains an important challenge which requires the use of several unit operations arranged in integrated system [6]. It related to the fact that fermentation broths have a complex composition [7]. Indeed, in addition to the main product, they contain water, residual substrates, microbial cells, and compounds originated from nutrients as well as carboxylic acids and other by-products. Another important issue that must always be considered is that compound fermentation broths may vary significantly with microbial species, activity, and type of substrate as well as fermentation conditions, such as pH and temperature [8,9,10].

The separation of fermentation broths can be performed by reverse osmosis (RO) technology. RO is a pressure-driven process, in which, when the applied pressure is greater than the osmotic pressure, the water molecules are forced to flow through a semipermeable membrane to the permeate side, while salts and low molecular weight organic materials are rejected [11,12,13] (Figure 1). As a result, the RO membrane separates the feed into two streams: pure water and a waste stream (named concentrate) [14,15]. It is well known that the separation is controlled mainly by the sieving mechanism [16], thus, it is strongly correlated with membrane molecular weight cut-off (MWCO) [17]. Generally, in the RO process, membranes with an MWCO of about 100 Da and a pore size below 1 nm are used [18,19,20]. More specifically, the RO process allows the rejection of uncharged organic compounds with molecular masses from 200 to 300 g/mol [16]. Nevertheless, the mechanism of separation of dissociated organic compounds through charged membrane differs. Indeed, when the solution pH is higher than the values of solute dissociation constant pKa, electrostatic repulsion may also have a significant impact on the separation process [21,22].

There is a general notion that treatment technology by RO outperforms the traditional processes. Indeed, RO is highly efficient technique which offers a simple operation and smaller floor space [23,24,25,26,27,28]. Moreover, as it has been pointed out by Kim [29], RO is relatively inexpensive to install, maintain, and operate. Hence, compared to the other desalination methods, RO is characterized by both the lowest energy demand and the lowest unit water cost [12]. It is worth noting that the RO superiority in terms of efficiency and cost over traditional methods means that currently, RO desalination installations comprise approximately 80% of all desalination plants in the world [30]. Until now, RO technology has been applied for desalination of seawater [31], brackish water [32], and wastewater treatment [33], as well as drinking water production from groundwater [34]. In addition, as it has been pointed out by Biesheuvel et al. [18], RO can be used to remove larger and uncharged molecules. Indeed, currently, there are new trends aimed at extending the range of RO. Several experimental studies have already investigated the use of RO membranes for separation of fermentation broths with the following main products: lactic acid [35,36,37], 2,3-butanediol [17,38], succinic acid [39,40], butyric acid [8], ethanol [41], and biohydrogen [42]. For instance, Davey et al. [17] studied the purification and concentration of a 2,3-butanediol fermentation broth with the use of nanofiltration (NF) and RO membranes. The authors noted that the polyamide RO membrane (MWCO = 100 g/mol) could be successfully used for the purification and concentration of the broth. Indeed, at pH of 6.5, the rejection of 2,3-butanediol and acetate was equal to 96.1% and 94.6%, respectively. Importantly, although a significant amount of research has been conducted in the application of reverse osmosis for various treatment processes, to the best of the authors’ knowledge, the separation of 1,3-PD fermentation broths by the RO technology has not been yet evaluated.

The first RO membrane was developed in the 1950s by Reid et al. [43]. RO technology advanced during the 1970s, mainly by the applications of hollow fiber membranes and the significant development of composite membranes [44]. Since 1980, polymeric NF and RO membranes have dominated the world market [45]. Generally, RO membranes have an ultrathin dense skin layer based on the polymeric support [8]. At present, the application of RO membranes fabricated from various polymers, such as cellulose acetate (CA), cellulose diacetate, polyamides, poly(arylene ether sulfone), piperazine, and chitosan have been studied [24,27,45]. Among them, CA and aromatic polyamides are the most widely applied material for fabrication of commercial RO membranes [46]. Vatanpou et al. [47] have clearly demonstrated that after 2009, the number of articles focused on the CA membranes increased significantly, which confirms the importance of CA in membrane production. Undoubtedly, this finding is related to the fact that CA membranes show a greater number of favourable chemical and materials properties than other membranes. Indeed, due to the presence of hydroxyl groups in the structure (Figure 2), CA membranes have a desirable hydrophilic property [47,48,49,50] and thus a potential affinity for water permeation [51]. Furthermore, they are biodegradable, biocompatible, easily manufactured, cheap, and can be easily derivatized [47,52,53,54]. However, cellulose acetate membranes undergo hydrolytic decomposition resulting in the substitution of hydroxyl groups by acetyl groups and a decline in the membrane performance [55].

It is noteworthy to mention that in the literature, studies on the use of reverse osmosis CA membranes for separation of fermentation broths are limited to lactic acid broths [36]. In view of the above, to the best of the authors’ knowledge, this is the first study aimed at investigating the feasibility of the cellulose acetate RO membrane for the separation of actual 1,3-PD broth. For this purpose, the installation with the plate membrane module has been used. Undoubtedly, results obtained by the present study will be particularly important for the designing of continuous membrane bioreactors used for production of 1,3-PD via glycerol fermentation process.

## 2. Results and Discussion

### 2.1. Membrane Performance

In general, water permeability and salt retention are the two main parameters used to determine membrane properties [26]. Hence, in the first step of the research presented, the relation between the water flux *J_w_* (permeate) and transmembrane pressure (TMP) was determined. As expected, the permeate flux increased with increasing TMP in a linear relation (Figure 3, “Start”). Indeed, when the pressure varied from 0.7 MPa to 1.0 MPa, the flux increased from 9.1 L/m^2^h to 13 L/m^2^h. A further increase in TMP to 1.2 MPa led to an increase in the membrane performance to 15.8 L/m^2^h. In other words, the obtained results confirmed that the water flux across the membrane is a linear function of the differential pressure between the feed pressure and the solution osmotic pressure [42]. This observation is in line with previous studies [8,17,56,57], wherein strong linear relationships between permeate flux and applied pressure for RO membranes were reported.

It is well known that the performance of RO membranes can be significantly diminished by the presence of solutes in the feed. Therefore, the membrane used was first tested with a model solution containing NaCl (2 g/L) and 1,3-PD (5 g/L). The obtained results showed that the permeate flux was lower than that reported for distilled water. Indeed, the flux of 11 L/m^2^h at operating pressure of 1 MPa has been noted (Figure 4, “NaCl + 1,3-PD”). According to the solution–diffusion model [58,59,60], it can be attributed to the decrease in TMP due to the increase in osmotic pressure Δπ by the presence of solutes, according to the following formula:(1)$$J_{w}=A_{w}\left(\Delta P-\Delta \pi \right),$$
where Δ*P* is the hydraulic pressure applied across the membrane and *A_w_* is the water permeability constant.

As the RO process proceeds, solutes rejected by the membrane form a layer near the membrane surface, which is known as the concentration polarization phenomenon. The accumulation of solutes is reversible and can be minimized by various methods, such as velocity adjustment, ultrasound, or an electric field [56]. Concentration polarization has been widely reported during the RO process of various types of feed, for instance NaCl-water solution [56] and raw water softened by caustic or lime [61].

It is worthy of note that the same order of magnitude of the permeate flux was recorded in [62], wherein the concentration of acrylic acid and acetic acid by the RO process was investigated. Indeed, in the aforementioned study, it was noted that during retention of acrylic (2.5%) and acetic (1.5%) acids, polyamide RO membranes provided the permeate flux of 7.89 L/m^2^h and 8.36 L/m^2^h, under pressure of 2 MPa and a temperature equal to 298 K.

As expected, more significant decline in the membrane performance was noted during the RO process of 1,3-PD fermentation broths (Figure 4, “S1“ and “S2“). The performed experiments have clearly demonstrated that during the broth separation the permeate flux was equal to 2 L/m^2^h, which constituted only 15% of its initial value. Undoubtedly, this finding was related to the fact that the feed contained, in addition to the solutes (Table 1), protein substances (yeast extract, meat extract, and peptone), which are known as the major organic foulants. The aforementioned compounds adsorbed and attached on the membrane surface, which led to the unfavorable formation of a cake layer along the membrane. As a result, a significant decrease in water passage was observed. These results are in good agreement with the work of Omwene et al. [39], where various RO membranes for the recovery of bio-based succinic acid were used. The aforementioned authors noted for new membranes the initial permeate flux of 10.6, 17.8, 18.3, and 8.1 L/m^2^h, at a pressure equal to 1.8 MPa. However, after 3 h of the treatment process run, average fluxes were reduced to a steady state of 6.1, 6.2, 6.3, and 7.1 L/m^2^h, respectively.

There is no doubt that investigation and understanding of the fouling phenomenon is of great importance since it affects membrane performance, permeate quality, and salt rejection [13]. The fouling of RO membranes has been widely described and discussed in several papers [20,63,64,65,66,67,68,69]. It should be pointed out that, since membrane fouling highly depends on the feed quality [63,70], one of the solutions for addressing this issue is the use of pre-treatment systems, allowing the removal of suspended solids and biological content from fermentation broths. Indeed, reliable and adequate pre-treatment of fermentation broths plays a critical role in ensuring their appropriate quality for RO technology and thus improving membranes performance. It has been widely documented that microorganisms and large particles can be successfully removed from fermentation broths by microfiltration (MF) [71,72,73] and ultrafiltration (UF) processes [6,23,74,75,76].

It is worthy of note that after about 60–90 min of the process run, the obtained permeate flux was stable over the course of the experiment (Figure 4, “S1” and “S2”). Surprisingly, it has been determined that rinsing of the membrane module with distilled water allowed the recovery of the initial process performance. As a result, despite the two runs (S1 and S2) of the 1,3-PD broth separation, the water flux determined after both series was equal to that reported for the new membrane (Figure 4, “water”). The importance of this point is that although CA membranes are generally characterized by low microbial resistance (since CA consists of β-dehydrated glucose, which is susceptible to bacterial attack [77]) the CA membrane used in the present study was resistant to irreversible fouling. This noteworthy result has a positive effect on the economic viability of the RO process. Indeed, impurities that cause irreversible fouling often lead to the requirement of membrane replacement or chemical cleaning, which, in turn, lead to increased process costs [78].

### 2.2. Separation of 1,3-Propanediol and Carboxylic Acids

The salt rejection rate refers to the membrane’s ability to separate solute [20], hence, many studies have paid attention to reporting various RO membranes characteristics in terms of the salt removal efficiency [26,36,41,61,79,80]. There is general agreement that RO membranes provide excellent and almost complete (mainly >99%) salt rejection. Nevertheless, according to the literature, the rejection characteristics is related to the membrane permeability [20] as well as the membrane effective charge, the solute nature, and concentration [79].

In turn, the results obtained in the present study have demonstrated that the presence of 1,3-PD (5 g/L) in the feed solution had a significant impact on the monovalent salt NaCl (2 g/L) retention by the CA membrane. Indeed, it has been observed (Figure 5) that the retention of salt ions was equal to 92%, which is much lower than the values reported in the literature. Furthermore, it has been determined that the 1,3-PD rejection was equal to about 40%.

The analysis of the fermentation broth composition has demonstrated that they consisted mainly of 1,3-PD and the following by-products: acetic acid, succinic acid, lactic acid, and formic acid. It is important to recognize that cellulose acetate is the uncharged polymer [28], hence, CA membranes have a neutral surface charge [50,81,82]. Therefore, it can be assumed that the separation mechanism in the present study was based mainly on the size effect. Thus, the component retention depended mainly on the membrane pore size and porosity, as well as the solute’s molecular weight and diffusion coefficient.

The objective of the performed RO process was to separate 1,3-PD from other broth components. The experiments performed on the real post-fermentation solutions have shown that 1,3-PD, which is a target product of the bioprocess, passed through the RO membrane (Figure 6a). Indeed, the 1,3-PD retention degree, for the first 50 min of the process runs, did not exceed 3% and systematically decreased. Finally, after 300 min, no retention of 1,3-PD was observed. Since the 1,3-PD retention from the model solution was equal to about 40% (Figure 5), its very high permeation across the membrane during the separation of actual fermentation broths was likely due to the variety of organic compounds present in the feed. The pH of fermentation broths was equal to 7. It is worth noting that since dissociation constant pKa of 1,3-PD is equal to 14.46 (Table 1), it can be assumed that it existed in the feed solution in molecular state.

The retention of main by-products of glycerol fermentation process (lactic acid, succinic acid, acetic acid, and formic acids) in the function of RO process time during two experimental runs is shown in Figure 6b. Since the pH control during fermentation was applied (adding NaOH solution, see Section 3.1), the carboxylic acids occurred in the broth also in the form of salts. It has been noted that acids were characterized by various retention degrees. Moreover, for each by-product a slight decline of retention was noted; however, the global rejection did not change significantly. For instance, the retention degree of succinic acid degreased from about 98% to 92%. In turn, for formic acid, a decline from 70% to 61% was noted. According to [39], the aforementioned decline in membrane rejection with an increase in recovery can be attributed to the reduction in the feed volume and the phenomenon of concentration polarization.

Succinic acid, a dicarboxylic acid with four carbon atoms, can exist in three forms: neutral, monovalent, and divalent [40]. In the present study, it was found that among all carboxylic acids present in the fermentation broths, succinic acid was characterized by the highest retention degree (92%). This can be explained by the fact that succinic acid existed as larger sized hydrated succinate anions (C2H4C2O42−) in the fermentation broth of a pH equal to 7.0 [39]:(CH_2_)_2_(CO_2_H)_2_ → (CH_2_)_2_ (CO_2_H) (CO^2^)^−^ + H^+^     pKa_1_ = 4.21,(2)
(3)$$({\mathrm{CH}}_{2}{)}_{2}\;({\mathrm{CO}}_{2}H)\;({\mathrm{CO}}_{2}{)}^{-}\;\to \;({\mathrm{CH}}_{2}{)}_{2}({\mathrm{CO}}_{2}{)}^{2-}H^{+}\;\;\;\;\;\mathrm{pKa}2_{\;}=5.64,$$
which promoted its strong rejection.

With regards to lactic acid, a dicarboxylic acid with four carbon atoms, the retention by the cellulose acetate RO membrane equal to 85% was noted. In turn, acetic acid, a monocarboxylic acid containing two carbon atoms, is an inhibitor of the growth of fermentation microorganisms in high concentration. In the present study, it was recognized as the major impurity of 1,3-PD fermentation broths. Indeed, it was noted that its concentration in was equal to about 3 g/L. Results obtained in the present study demonstrated that the cellulose acetate RO membrane provided acetic acid retention of about 70%. Finally, formic acid, the simplest carboxylic acid, containing a single carbon, demonstrates the smallest structure in terms of molecular weight in the present study. Accordingly, for this by-product, the used membrane showed the lowest retention degree, equal to about 61%.

The retention degree in the function of acids’ molecular weight is presented in Figure 7. It can be observed that the acid retention tends to increase with increasing molecular weight, according to the following order: succinic acid > lactic acid > acetic acid > formic acid. This finding is in agreement with previous work published in the literature [83], wherein the rejection of acetic acid by a polyamide membrane was investigated. In the aforementioned study, it was noted that for carboxylic acids rejection increases with increasing molecular mass. Likewise, Diltz et al. [42] reported the positive correlation of rejection degree with the molecular weights of components of simulated biohydrogen broth. Finally, it is essential to mention that in the present study, no relation between acid pKa and retention degree has been found.

### 2.3. Separation of Salts Anions and Cations

The analysis of the fermentation broths composition indicated the presence in the feed of the following anions: Cl^−^, PO_4_^3−^, SO_4_^2−^, NO_3_^−^, and cations as follows: K^+^, Na^+^, Ca^2+^, and Mg^2+^ (Table 2). The retention degrees of cations and anions during the separation of broths are presented in Figure 8. It can be clearly observed that the used membrane ensured stable ion separation efficiency. Notwithstanding, various reject degrees have been noted. Indeed, divalent (SO_4_^2−^) and trivalent anions (PO_4_^3−^) were completely retained by the membrane, whereas the rejection degrees of monovalent anions NO_3_^−^ and Cl^−^ were significantly lower and equal to about 80% and 40%, respectively (Figure 8a). More specifically, the order of anions rejection was SO_4_^2−^ > PO_4_^3−^ > NO_3_^−^ > Cl^−^. According to Epsztein et al. [82] low rejections noted for Cl^−^ and NO_3_^−^ can be attributed to the absence of the Donnan-exclusion mechanism with the CA membrane.

With regards to cations, the highest retention was noted for Ca^2+^ (88%), and the lowest for NH_4_^+^ (54%) (Figure 8b). The following order of cations retention degree was determined: Ca^2+^ > Na^+^ > K^+^ > NH_4_^+^. Therefore, it can be clearly indicated that the highest retention degrees for divalent and trivalent ions were recorded. These results are similar to those reported in the work of Phanthumchinda et al. [37], wherein polyamide thin-film composite RO membranes were used to recover and purify lactic acid from fermentation broth. The aforementioned authors reported various retention degrees of ions present in the feed. For instance, it was pointed out that Ca^2+^ was retained, meanwhile Na^+^ and NH_4_^+^ passed through the negatively charged membrane. Based on the findings presented above, it can be summated that the retention degree with the cellulose acetate RO membrane increases with the increase of ionic radius and decrease of diffusion coefficient.

In the RO process, ion flux across the membrane is generated by the chemical potential gradient, such as concentration and electrical gradients [84]. More specifically, species that migrate through the membrane flow into the permeate side first, dissolve in the membrane material, and subsequently molecularly diffuse through the membrane active layer [44]. Hence, according to Mukherjee and Sengupta [85] permeation rates of ionized electrolytes during the RO process may be determined based on their ionic radius and diffusion coefficient data [85]. It is worthy of note that Lonsdale et al. [80] have pointed out that cellulose acetate membranes demonstrate a very high degree of ions rejection, which is not strongly sensitive to salt concentration. Taking the abovementioned into account, in the present study, the correlation between the retention degree of ions (anions and cations) present in 1,3-PD fermentation broths and their hydrated radius and diffusion coefficients has been investigated. As indicated above, the highest retention degrees (≥90%) for SO_4_^2^, PO_4_^3−^, and Ca^2+^ were noted. According to the results presented in Figure 9, it can be explained by the fact that the aforementioned ions are characterized by the highest ion radius and the lowest diffusion coefficients. Indeed, the hydrated radius and diffusion coefficients of SO_4_^2^, PO_4_^3−^ and Ca^2^ are equal to 0.379 nm, 0.339 nm, 0.412 nm and 1.07·10^−9^ m^2^/s, 0.612·10^−9^ m^2^/s, 0.790·10^−9^ m^2^/s, respectively. Contrary, the lowest retention (<50%) was recorded for Cl^−^. Undoubtedly, this is related to the fact that anion Cl^−^ is characterized by the smallest hydrated radius and the highest diffusion coefficient, equal to 0.332 nm and 2.03·10^−9^ m^2^/s, respectively.

The results obtained in the present study demonstrate that the used membrane ensured: (i) the migration of 1,3-PD to the permeate side, (ii) a retention degree higher than 50% of all by-products and (iii) recycling of nutrients in the significant amount (varying from 40 to 100%). It clearly indicates the possibility for the application of continuous membrane bioreactors coupled with cellulose acetate RO membranes for production of 1,3-propanediol via glycerol fermentation. In the subsequent steps (in order to obtain pure components present in the permeate) additional operation, such as an ion exchange, is required.

## 3. Materials and Methods

### 3.1. Glycerol Fermentation

In the present study, as a feed, the real glycerol post-fermentation solution was used. The glycerol fermentation was carried out using *Citrobacter freundii* bacteria (isolated and characterized in the Department of Biotechnology and Food Microbiology, Poznań University of Life Science, Poland). The process was carried out in a LiFlusGX bioreactor (Biotron Inc., Seoul, Korea) at temperature 304 K. A prepared broth contained per liter: glycerol (30.0 g), yeast extract (2.0 g), meat extract (1.5 g), peptone K (2.5 g), K_2_HPO_4_·3H_2_O (3.4 g), KH_2_PO_4_ (1.3 g), MgSO_4_·7H_2_O (0.4 g), (NH_4_)_2_SO_4_ (2.0 g), CaCl_2_·2H_2_O (0.1 g), and CoCl_2_·6H_2_O (0.004 g). After sterilization, the medium was inoculated with bacteria in a lag phase (10% *v*/*v*) and the batch fermentation was performed using a bioreactor (LiFlusGX, Biotron Inc., Seoul, Korea) at 300 K for 2 days. The pH value was maintained at 7.0 (±0.2) by automatic additions of 5 M NaOH. The conditions of the fermentation process are presented in more detail in previous works [89,90].

The composition of the obtained post-fermentation solutions is presented in Table 1 and Table 2. The fermentation processes was performed ten times. In the paper, the results of two series of experimental runs were reported (Series 1 and Series 2). After completing the fermentation process, the solution was pretreated by gravitational sedimentation for four hours. Subsequently, it were used as a feed for the treatment by the RO process (Figure 10). In the absence of sedimentation, high turbidity of the fermentation broths could lead to intense fouling phenomenon, resulting in low membrane performance, frequent membrane cleaning, and low permeate quality.

### 3.2. RO Set-Up

Reverse osmosis experiments were carried out using the system represented in Figure 11. The installation was equipped with the flat SEPA-CFII membrane module manufactured by GE Osmonics (Minnetonka, MN, USA). The commercially available Sepa CF cellulose acetate RO CE membranes was used. The membrane active area was equal to 150 cm^2^. The determined water contact angle (WCA) was 55.4 ± 5.8°.

The experiments were conducted under the temperature of 295 (±1) K. The process temperature was stabilized by a cooling system (thermostatic Grundfos valve). The transmembrane pressure was equal to 1 MPa for all runs. The feed was taken from the tank (volume 5 L) by a piston pump. Retentate and obtained permeate were returned to the feed tank, which allowed the maintenance of constant feed concentration during the tests.

RO process performance was determined in terms of permeate flux, which is described as the volume of permeate obtained over a defined period of time, according to the following formula:(4)$$J=\frac{dV}{dt\cdot S},$$
where d*V* is permeate cumulative volume (L), *S* is the total active membrane area (m^2^) and *t* is time (h).

The membrane retention *R* (%) of each solute was obtained was calculated as follows:(5)$$R=\left[\frac{\left(C_{F}-C_{P}\right)}{C_{F}}\right]\cdot 100\%,$$
where *C**_F_* and *C**_P_* are the solute concentration (g/L) in the feed and the permeate, respectively. Hence, *R* equal to 100% indicates complete solute rejection, while *R* of 0 no separation [17].

### 3.3. Analytical Methods

The membrane hydrophobicity was determined by WCA using the Contact Angle System OCA (Data Physics, Filderstadt, Germany) apparatus.

The concentrations of 1,3-PD, glycerol and the organic acids in the feed and obtained permeate were determined by high-performance liquid chromatography HPLC using a UlitiMate 3000 (Thermo Fisher Scientific, Germering, Germany) with refractometer detector R1-101 Shodex (Showa Denko America, New York, NY, USA) and a column Aminex HPX-87H (BIO RAD, Berkeley, CA, USA) with HyperREZ XP H+ Guard (Thermo Scientific, Waltham, MA, USA), through which a H_2_SO_4_ solution (0.005 M) flowed (0.6 mL/min).

The anion and cation concentrations were determined using an ion chromatograph (Herisau Metrohm AG, Herisau, Switzerland). The separation of anions was achieved on a 1.7 mm × 3.5 mm Metrosep RP guard column in series with a 250 mm × 4.0 mm Metrohm A Supp5-250 analytical column. An analytical column 150 mm × 4.0 mm Metrosep C2-150 was used for the separation of cations.

The pH was measured by using the multifunctional ULTRAMETER 6P meter (Myron L Company, Carlsbad, CA, USA).

## 4. Conclusions

The results obtained in the present study highlight the efficiency of the reverse osmosis process in the treatment of biological solutions. Indeed, the experimental data presented establish the technical feasibility of using a cellulose acetate RO membrane for the separation of 1,3-propanediol fermentation broths. It has been found that 1,3-PD, which is as a target product, passed through the membrane into the permeate side, meanwhile other broth components were rejected in various degrees. It has been shown that retention of carboxylic acids tended to increase with increasing molecular weight, according to the following order: succinic acid > lactic acid > acetic acid > formic acid. In turn, with regards to ions, it has been determined that retention increased with the inctrease of ionic radius and the decrease of the diffusion coefficient. Furthermore, the results obtained in the present study have shown that the rinsing of the membrane module with distilled water allowed recovery of the initial process performance. This noteworthy result indicates that CA membranes demonstrated resistance to irreversible fouling, which positively effects the economic process viability. Finally, it should be pointed out that the main outcome of the present study may be applied for the designing of continuous membrane bioreactors combined with the cellulose acetate RO membrane used for the production of 1,3-propanediol via glycerol fermentation process.

Despite the present study demonstrating the application of cellulose acetate RO materials for the treatment of broths, there is a substantial potential for future research. Indeed, a more detailed study would examine other aspects like the impact of feed pH and process conditions, such as temperature and applied pressure, on the CA membrane performance and efficiency.

## Figures and Tables

**Figure 1 ijms-23-11738-f001:**
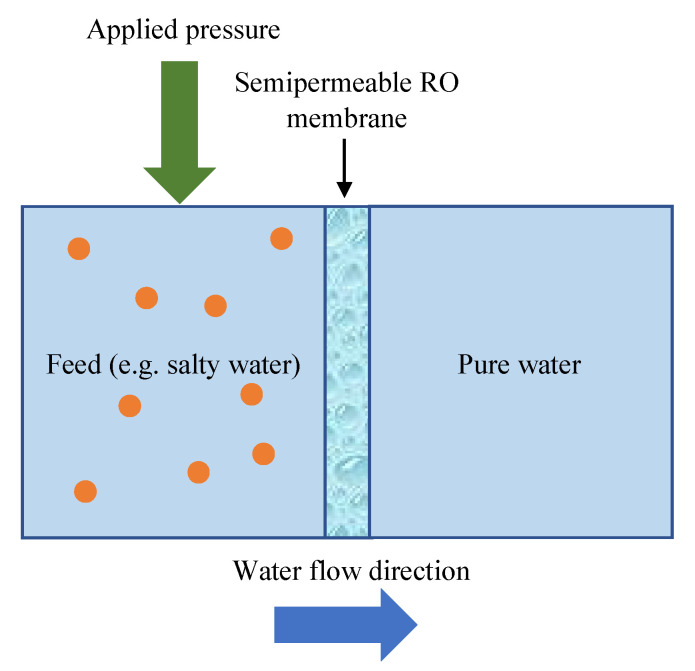
Schematic diagram of RO process.

**Figure 2 ijms-23-11738-f002:**
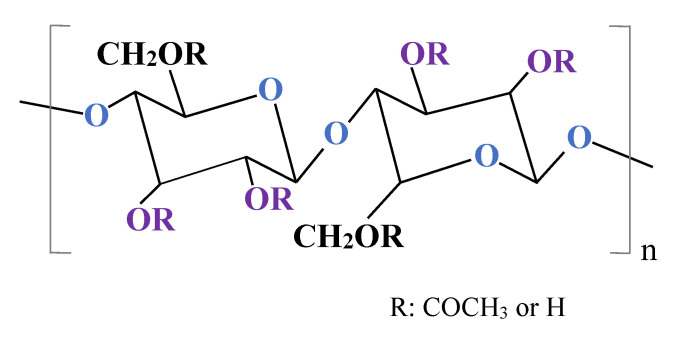
Chemical structure of cellulose acetate.

**Figure 3 ijms-23-11738-f003:**
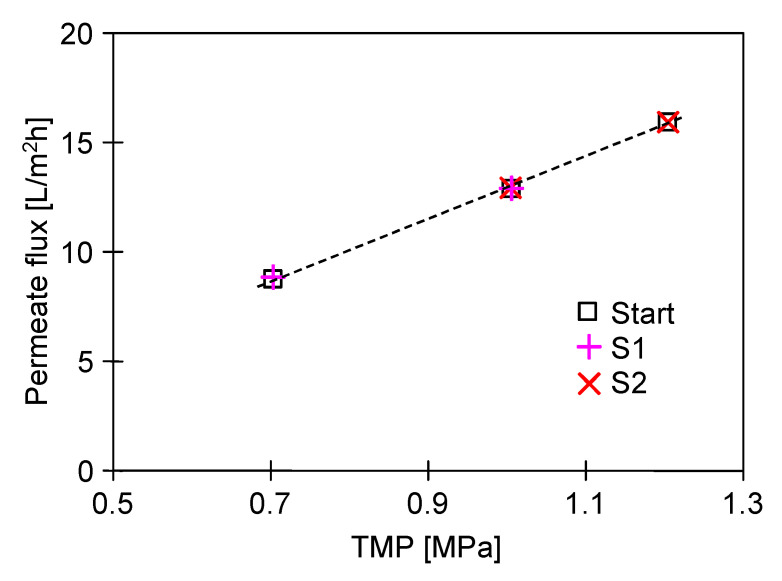
The impact of TMP on the permeate flux. Start-water flux before the broth separation process, S1 and S2-water flux after the broth separation process (series S1 and S2, respectively).

**Figure 4 ijms-23-11738-f004:**
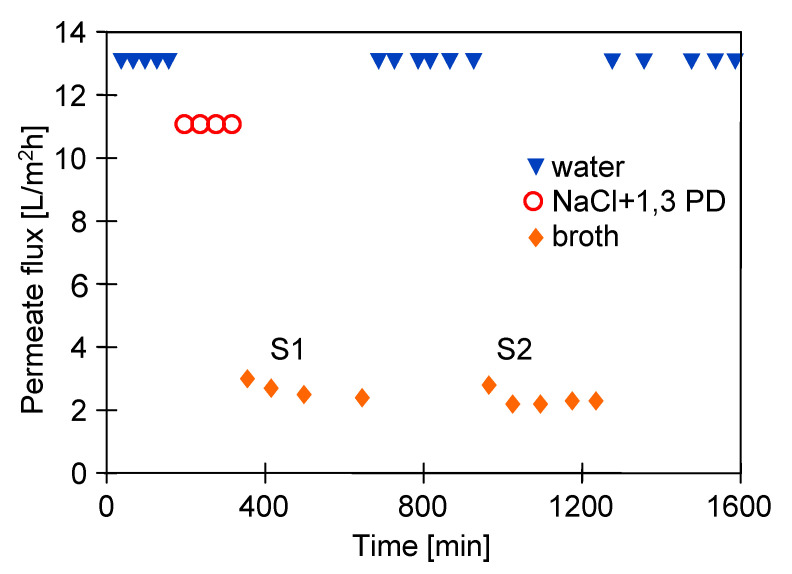
Impact of feed composition on the permeate flux. Feed: distilled water (“water”), solution of NaCl (2 g/L) and 1,3-PD (5 g/L) (“NaCl + 1,3-PD”), fermentation broth (“broth”). TMP = 1 MPa.

**Figure 5 ijms-23-11738-f005:**
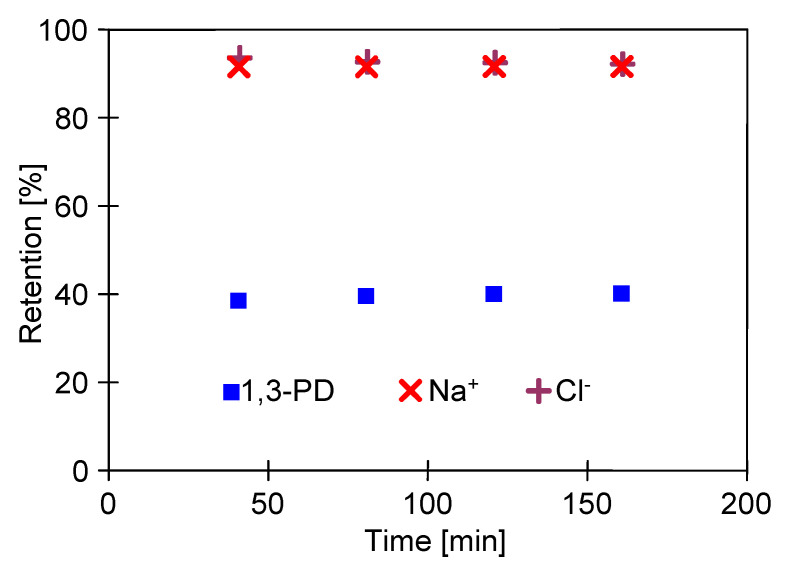
Changes in the retention degree of 1,3-PD and NaCl. Feed: model solution of 1,3-PD (5 g/L) and NaCl (2 g/L), TMP = 1 MPa.

**Figure 6 ijms-23-11738-f006:**
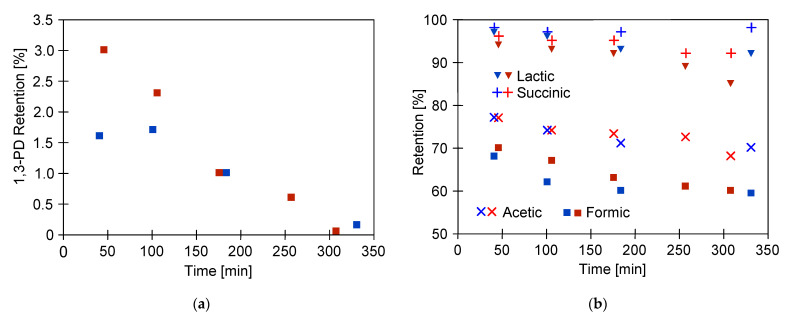
Changes in the retention degrees during separation of fermentation broth: (**a**) 1,3-propanediol; (**b**) carboxylic acids. TMP = 1 MPa, blue symbol—Series 1, red symbol—Series 2.

**Figure 7 ijms-23-11738-f007:**
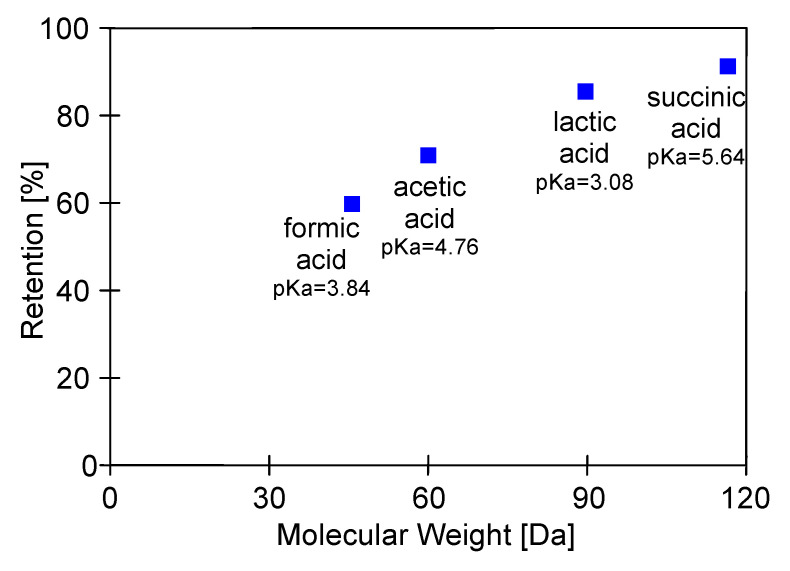
Retention as a function of molecular weight.

**Figure 8 ijms-23-11738-f008:**
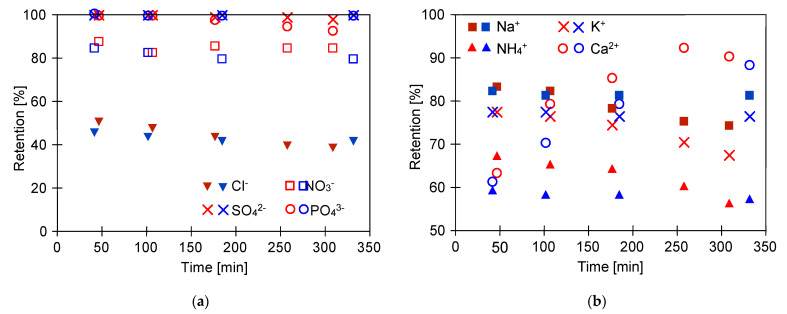
Changes in the retention degrees during separation of fermentation broth: (**a**) Anions; (**b**) cations. TMP = 1 MPa, blue symbol—Series 1, red symbol—Series 2.

**Figure 9 ijms-23-11738-f009:**
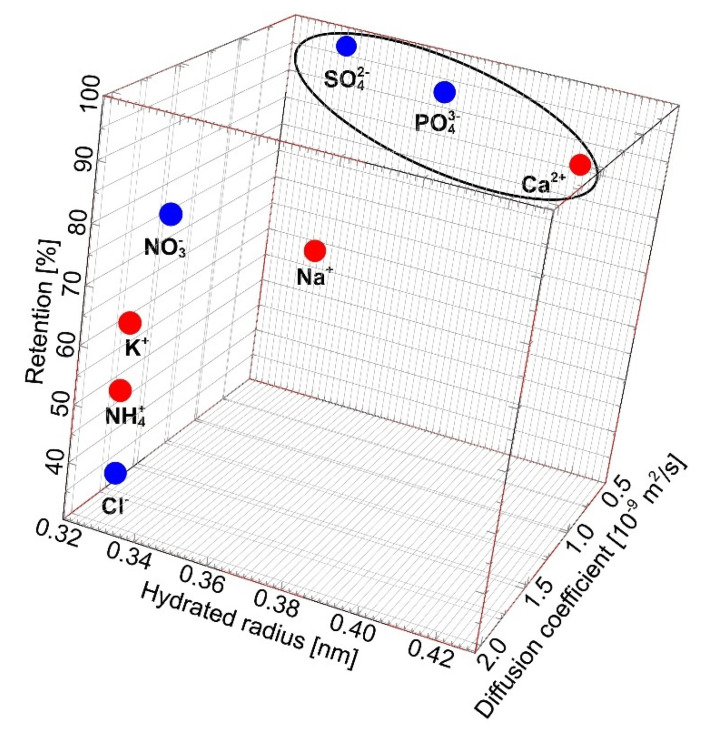
Ion retention as a function of ions hydrated radius and diffusion coefficient. Hydrates radius and diffusion coefficient: data from [86,87,88].

**Figure 10 ijms-23-11738-f010:**
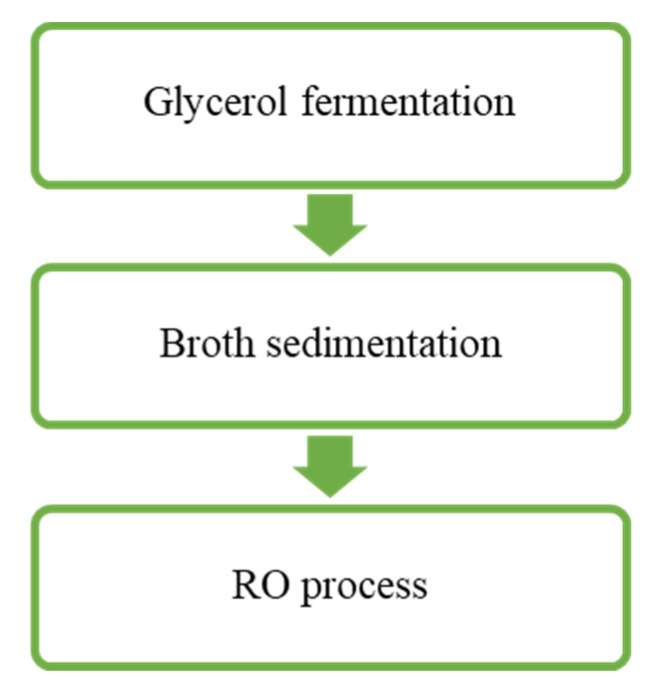
Flow diagram of process performed in the present study.

**Figure 11 ijms-23-11738-f011:**
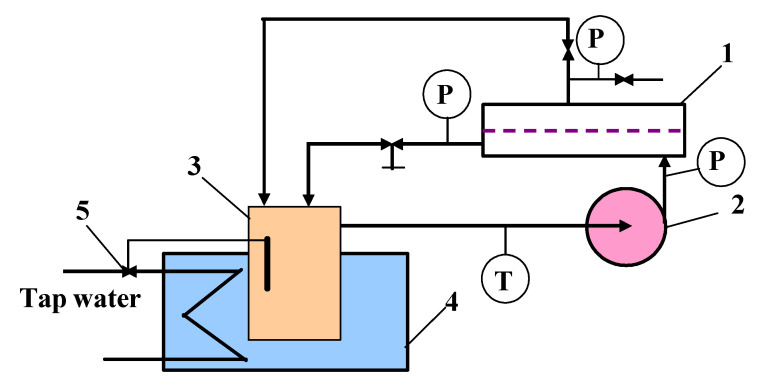
Experimental RO set-up. 1—SEPA-CFII RO module, 2—pump, 3—feed tank, 4—cooling bath, 5—thermostatic Grundfos valve, P—manometer, T—thermometer.

**Table 1 ijms-23-11738-t001:** Characteristics and concentrations of components present in 1,3-PD fermentation broths.

Component	Molecular Formula	Molecular Weight [Da]	Dissociation Constant pKa	Concentration [g/L]	
				Series 1	Series 2
1,3-propanediol	C_3_H_8_O_2_	76.09	14.46	12.65	11.52
acetic acid	C_2_H_4_O_2_	60.05	4.76	2.98	3.13
succinic acid	C_4_H_6_O_4_	118.08	4.21 and 5.64	1.35	1.45
lactic acid	C_3_H_6_O_3_	90.08	3.08	0.78	0.63
formic acid	CH_2_O_2_	46.05	3.84	0.68	0.56
glycerol	C_3_H_8_O_3_	92.09	14.40	0.08	0.62

**Table 2 ijms-23-11738-t002:** Characteristics and concentrations of ions present in 1,3-PD fermentation broths.

Ion	Hydrated Radius [nm] ^1^	Diffusion Coefficient [m^2^/s] ^1^	Concentration [g/L]	
			Series 1	Series 2
Cl^−^	0.332	2.03	1.34	1.92
PO_4_^3−^	0.339	0.61	2.56	2.67
SO_4_^2−^	0.379	1.07	1.86	1.58
NH_4_^+^	0.331	1.98	0.62	0.65
K^+^	0.331	1.96	1.82	1.94
Na^+^	0.358	1.33	3.85	3.94
Ca^2+^	0.412	0.79	0.05	0.06
Mg^2+^	0.428	0.71	0.01	0.01

^1^ Data from [86,87,88].

## Data Availability

Not applicable.

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
