# Peer review of "The Application of Cellulose Acetate Membranes for Separation of Fermentation Broths by the Reverse Osmosis: A Feasibility Study"

_ijms, 2022, doi:10.3390/ijms231911738_

Round 1
Reviewer 1 Report
Q1. P2 Line 72, “fermentation broth with the use of NF and RO membranes”. Acronyms/Abbreviations/Initialisms should be defined the first time they appear in each of three sections: the abstract; the main text; the first figure or table. When defined for the first time, the acronym/abbreviation/initialism should be added in parentheses after the written-out form.
Q2. P2 Line 71-72, “For instance, Davey et al. [16] 70 have studied the purification and concentration of a 2,3-butanediol producing gas fermentation broth with the use of NF and RO membranes”. The 2,3-butanediol is produced in the liquid phase, not gas
Q3. Check the unit consistency
Q4. Page 4 Line 125, 1 MPa has been noted (Figure 3, 125 “NaCl+1,3-PD”). The wrong citation; it should be Figure 4
Q5. Page 4 Line 173, reported for the new membrane (Figure 3, “water“). The wrong citation; it should be Figure 4. Carefully recheck throughout the manuscript
Q6. P 12 Line 367, “was calculated calculated” duplicates words
Q7. Materials and methods. Why did the author use only pure glycerol as the carbon source? In practice, crude glycerol is commonly used to produce 1,3-PDO due to its waste and low cost.
Q8. P 10 Line 336 The composition of the obtained post-fermentation solutions is presented in Tables 1 and 2. The fermentation processes were performed several times. It is useful to clarify what several times.
Q9. After completing the fermentation process, the solution was pretreated by the conventional sedimentation for several hours. It is useful to clarify what several hours.
Q10. After completing the fermentation process, the solution was pretreated by the conventional sedimentation for several hours.
It is useful to clarify what conventational sedimentation gravitation or ?.
Q11. Why did the author pick up only series 1 and 2 to analyze the data? How round did the author test?
Q12. As mentioned in Q7, crude glycerol generally contains impurities especially salt and free fatty acids, that might affect the purification by RO. Did the author concerned about that point?
Q13. As the author mentioned in the introduction, RO is a well-known and easy to use for purified 1,3-PDO than other methods.
I suggest the author add the results compared with other techniques in terms of efficiency and cost.
Author Response
Dear Reviewer,
Dear Reviewer,
We would like to express our great appreciation to you for your interest in our work and the valuable comments and constructive suggestions on our manuscript. As indicated below, we have taken into considerations all your comments and we have made changes and corrections accordingly to your indications.
All changes made to the manuscript are highlighted in blue.
Thank you for your time and effort.
Yours sincerely,
Wirginia Tomczak and Marek Gryta
Q1. P2 Line 72, “fermentation broth with the use of NF and RO membranes”. Acronyms/Abbreviations/Initialisms should be defined the first time they appear in each of three sections: the abstract; the main text; the first figure or table. When defined for the first time, the acronym/abbreviation/initialism should be added in parentheses after the written-out form.
Thank you very much. We corrected this mistake.
Q2. P2 Line 71-72, “For instance, Davey et al. [16] 70 have studied the purification and concentration of a 2,3-butanediol producing gas fermentation broth with the use of NF and RO membranes”. The 2,3-butanediol is produced in the liquid phase, not gas
Thank you very much. We corrected this mistake as follows: “Davey et al. [16] 70 have studied the purification and concentration of a 2,3-butanediol fermentation broth…”
Q3. Check the unit consistency
Thank you very much. We checked the unit consistency.
Q4. Page 4 Line 125, 1 MPa has been noted (Figure 3, 125 “NaCl+1,3-PD”). The wrong citation; it should be Figure 4
Thank you very much. We corrected this mistake.
Q5. Page 4 Line 173, reported for the new membrane (Figure 3, “water“). The wrong citation; it should be Figure 4. Carefully recheck throughout the manuscript
Thank you very much. We corrected this mistake.
Q6. P 12 Line 367, “was calculated calculated” duplicates words
Thank you very much. We corrected this mistake.
Q7. Materials and methods. Why did the author use only pure glycerol as the carbon source? In practice, crude glycerol is commonly used to produce 1,3-PDO due to its waste and low cost.
We agree with the Reviewer's comment. Nevertheless, we want to emphasize that the presented studies were preliminary studies aimed at investigating how the impact of the broth components on the separation during the RO process.
Q8. P 10 Line 336 The composition of the obtained post-fermentation solutions is presented in Tables 1 and 2. The fermentation processes were performed several times. It is useful to clarify what several times.
We agree with the Reviewer's comment. We corrected this mistake and we wrote that the fermentation processes were performed ten times, as follows:
“The fermentation processes were performed ten times.”
Q9. After completing the fermentation process, the solution was pretreated by the conventional sedimentation for several hours. It is useful to clarify what several hours.
We agree with the Reviewer's comment. We corrected this mistake and we wrote that the fermentation processes were performed ten times, as follows:
“After completing the fermentation process, the solution was pretreated by the gravitational sedimentation for four hours.”
Q10. After completing the fermentation process, the solution was pretreated by the conventional sedimentation for several hours.
It is useful to clarify what conventational sedimentation gravitation or ?.
We agree with the Reviewer's comment. We corrected this mistake and we wrote that the fermentation processes were performed ten times, as follows:
“After completing the fermentation process, the solution was pretreated by the gravitational sedimentation for four hours.”
Q11. Why did the author pick up only series 1 and 2 to analyze the data? How round did the author test?
We decided to present only two series of the results obtained in order to ensure the transparency of the results. The concentrations of the broth components were very similar, without any deviations, therefore it was not necessary to present the results of all measurement series.
Q12. As mentioned in Q7, crude glycerol generally contains impurities especially salt and free fatty acids, that might affect the purification by RO. Did the author concerned about that point?
We agree with the Reviewer’s comment that the crude glycerol may contain impurities that would affect the separation process. However, as we indicated in the answer for Q7, the presented studies were preliminary studies aimed at investigating the impact of the broth components on the separation during the RO process. The further studies with the use of crude glycerol would be undoubtedly interesting.
Q13. As the author mentioned in the introduction, RO is a well-known and easy to use for purified 1,3-PDO than other methods.
I suggest the author add the results compared with other techniques in terms of efficiency and cost.
According to the Reviewer comment, we added the following information (lines 60-68):
There is a general notion that the treatment technology by RO has been outperforming over the traditional processes. Indeed, RO is highly efficient technique which offers simple operation and smaller floor space [23–28]. Moreover, as it has been pointed out by Kim [29], RO is relatively inexpensive to install, maintain and operate. Hence, compared to the other desalination methods, RO is characterized by both the lowest energy demand and the lowest unit water cost [12]. It is worth noting that the RO superiority in terms of efficiency and cost over traditional methods caused that nowadays, RO desalination installations comprise approximately 80% of all desalination plants in the world [30].
Best regards,
Wirginia Tomczak and Marek Gryta
Reviewer 2 Report
The review manuscript titled “The Application of Cellulose Acetate Membranes for Separation of Fermentation Broths by the Reverse Osmosis: A Feasibility Study” and written by Wirginia Tomczak and Marek Gryta could be interesting, but the section material and methods should be section 2 instead of 3. In general, the paper is well written, and the references are appropriate. I recommend a minor revision based on the following comments:
1. The section material and methods should be the section 2 instead of section 3.
2. Page 4, lines 134-136. The authors should include the impact of feed spacer geometries on concentration polarization phenomena in spiral wound membrane modules. I recommend including the following studies: Water 11 (1), 152; Separation and Purification Technology 192, pp. 441-456; Processes 8 (6), 692.
3. Could the authors provide a picture of the experimental setup?
4. How was measure the TMP exactly? Usually feed pressure and the pressure in the rejection line are measured.
Author Response
Dear Reviewer,
We would like to express our great appreciation to you for your interest in our work and the valuable comments and constructive suggestions on our manuscript. As indicated below, we have taken into considerations all your comments and we have made changes and corrections accordingly to your indications.
All changes made to the manuscript are highlighted in blue.
Thank you for your time and effort.
Yours sincerely,
Wirginia Tomczak and Marek Gryta
The review manuscript titled “The Application of Cellulose Acetate Membranes for Separation of Fermentation Broths by the Reverse Osmosis: A Feasibility Study” and written by Wirginia Tomczak and Marek Gryta could be interesting, but the section material and methods should be section 2 instead of 3. In general, the paper is well written, and the references are appropriate. I recommend a minor revision based on the following comments:
- The section material and methods should be the section 2 instead of section 3.
Thank you for this comment, however, following the article template for this journal, methods should be presented in Section 3.
- Page 4, lines 134-136. The authors should include the impact of feed spacer geometries on concentration polarization phenomena in spiral wound membrane modules. I recommend including the following studies: Water 11 (1), 152; Separation and Purification Technology 192, pp. 441-456; Processes 8 (6), 692.
We increased the citations for two papers indicated by the Reviewer, as follows:
The fouling of RO membranes has been widely described and discussed in several papers [20,63–69].
- Ruiz-García, A.; Nuez, I. Performance Assessment of SWRO Spiral-Wound Membrane Modules with Different Feed Spacer Dimensions. Processes 2020, 8, 692, doi:10.3390/pr8060692.
- Ruiz-García, A.; de la Nuez Pestana, I. Feed Spacer Geometries and Permeability Coefficients. Effect on the Performance in BWRO Spriral-Wound Membrane Modules. Water 2019, 11, 152, doi:10.3390/w11010152.
- Could the authors provide a picture of the experimental setup?
We are sorry, unfortunately, we are unable to provide a picture of the experimental setup. Currently, the installation is no longer in our laboratory.
- How was measure the TMP exactly? Usually feed pressure and the pressure in the rejection line are measured.
Thank you very much for this comment. We corrected the Figure 13 (Experimental RO set-up). The installation was equipped with three pressure gauges:
- feed line: inlet (p1) and outlet (p2),
- permeate side (p3).
The TMP was determined as follows: TMP=(p1-p2)/2-p3
.
Best regards,
Wirginia Tomczak and Marek Gryta
Round 2
Reviewer 1 Report
All comments are responsed.